**Data Availability Statement:** The de-identifiable data of the qualitative section for thematic analysis and the quantitative data for SEM-PLS analysis are

# Eco-friendly fashion among generation Z: Mixed-methods study on price value image, customer fulfillment, and pro-environmental behavior

**Khoa Tran**[1], **Tuyet Nguyen**[1,2]*, **Yen Tran**[1], **Anh Nguyen**[1], **Khang Luu**[1], **Y. Nguyen**[1]

**1** Youth Lab for Social Innovation, Ho Chi Minh City, Vietnam, **2** Department of Business, Minerva University, San Francisco, California, United States of America

* tuyet.nguyen@uni.minerva.edu

## Abstract

Raising environmental awareness and product development are two separate and costly investments that many small and medium-sized fashion businesses cannot afford to achieve sustainability. Therefore, there is a need to determine which factors exert a more significant impact on consumer loyalty and purchase intention toward eco-friendly fashions. Thus, this study employs a mixed-methods approach with thematic analysis and the SEM-PLS technique to research how Vietnamese Gen Z's perceptions of product-service quality, environmental awareness, and pro-environmental behavior influence their purchase intention and loyalty toward eco-friendly fashion products. Most interviewees acknowledged that they primarily gained knowledge about eco-friendly fashion through social media platforms. The qualitative results further showed that their knowledge of and attitudes toward eco-friendly fashion practices were insufficient to convince young customers to afford eco-friendly fashion products. The SEM-PLS results of 313 participants show that while customers' perceived behavioral control plays a more significant role in stimulating purchase intention, only product-service quality factors impact loyalty. Hence, this study suggests that businesses should prioritize improving service and product quality rather than funding green marketing when targeting Vietnamese Gen Z in case of financial constraints. Government should prioritize financial and technological support for fashion firms to develop high-quality eco-friendly fashion to ensure the product availability.

## Introduction

### 1. Sustainability in fashion

Sustainability is commonly referred to as a development that fulfills current demands and simultaneously ensures the capabilities of future generations to satisfy their needs [1]. This concept aims to guarantee intergenerational resource security. Within businesses, this philosophy is also applied to indicate the capacity of companies to address their present financial

available publicly without restriction at: https://doi.org/10.6084/m9.figshare.17021879.v3. The authors want to inform that one interviewee does not agree to record and transcribe their session; hence, only 23 transcription files are available. The original language of the qualitative data is Vietnamese.

**Funding:** The authors received no specific funding for this work.

**Competing interests:** The authors have declared that no competing interests exist.

objectives without jeopardizing incentives to achieve their goals in the long run [2]. However, current global clothing manufacturing is not sustainable. For instance, cotton cultivation requires high water intensity and insecticides whereas polyester, a synthetic material, is produced from unstainable oil [3, 4]. Manufacturers also discharge untreated dye effluents into local water systems. The consequences include higher concentrations of toxicants and heavy metals in the water body and harm to the health of aquatic animals and nearby residents [5]. Furthermore, the rise in fashion purchases has resulted in a new phenomenon of disposing of garments to follow new fashion trends despite the quality and durability of products. This large amount of textile waste results in substantial environmental degradation.

In contrast, eco-friendly or green fashion refers to clothing and accessories produced with minimal chemicals, pesticides, or toxic pigments which significantly lower the ecological footprint. Hence, sustainable fashion is in high demand to protect the environment in the long term. Businesses worldwide have integrated sustainability aspects in their fashion products and gained different benefits in return. For instance, Kotn, a Canadian fashion brand that specializes in sustainable fashion, received customer support and quickly expanded its market into an international one because of its eco-friendly nature. More precisely, since it was first introduced in 2015, the brand has achieved an average of 37% month-over-month growth [6]. Another example is Patagonia, a fashion business with a mission to promote eco-friendly fashion. With its environmental campaigns, the annual sales of $1 billion have made this company a large corporation [7]. Owing to the increased availability and options to purchase eco-friendly fashion, consumers are also becoming familiar with products having elements of recycling, second-hand and natural fiber materials, and environmentally friendly fabrics [8, 9]. However, the achievements of Kohn and Patagonia are outliers rather than illustrating the success of the industry.

Scholars have shown the need for collaboration between suppliers and consumers to promote eco-friendly fashions. More specifically, despite consumer awareness and advocacy toward environmental measures, their actions generally do not reflect their knowledge and attitudes [10]. Additionally, recent studies have recognized the lack of research on how after-sales fulfillment and price value image influence people's decisions to buy eco-fashion products and their repurchase intentions as the literature tends to examine customers' environmental concerns and knowledge as a factor rather than customers' perceived satisfaction with product and service quality [11]. In short, they concentrated only on the promotional elements of the marketing mix (4P). Therefore, rather than only investigating environmental concerns, environmental awareness and perception as in previous Theory of Reasoned Action (TRA) and Theory of Planned Behavior (TPB) studies [12], the literature needs to adopt a second set of constructs illustrating the product & service qualities and price value of eco-friendly fashion products, addressing the two additional marketing elements (Product and Price). From a practical perspective, rising environmental awareness and product development are two separate and costly investments that many small and medium-sized businesses cannot afford. Therefore, the literature gap motivates researchers to determine which factors have a stronger impact on consumer loyalty and purchase intention toward eco-friendly fashion: consumers' understanding of environmental preservation or the development of quality eco-friendly products and services? This research question provides the foundation for constructing a conceptual framework and research hypotheses.

## 2. Vietnamese fashion industry and generation Z's attitudes

Vietnam is an important context for eco-friendly studies because failure to pursue eco-friendly practices affects millions of workers in the domestic market and hinders the global supply

chain. From a domestic perspective, the garment and textile industry is one of the key industries in Vietnam, accounting for the country's second-largest export turnover and contributing 16 percent of the total GDP in 2019 [13]. Simultaneously, the textile and garment industry employs over 1.6 million people, more than 12% of the industrial workforce, and almost 5%, of the total labor force in the country [14]. However, the textile industry faces various environmental issues [15]. Chemical processing, such as dyeing and printing, accounts for 20% of Vietnamese water pollution [16]. Hence, Vietnam is facing a trade-off between economic development and the environment, long-term health, and social well-being. In addition, as foreign importers and global fashion firms have raised their higher standards of sustainability in fashion production, failure to comply with green production would result in the loss of jobs for millions of citizens and multi-billion dollars in GDP [15, 16]. From the international perspective, according to the World Trade Organization, Vietnam ranked top three global apparel exporters in 2020 [17]. Therefore, while the peer-reviewed research on the Vietnamese fashion and garment industry is still limited, various international media and international corporate reports have highlighted the significance of Vietnam in the global supply chain. A halt in Vietnamese garment manufacturing due to environmental concerns could disrupt the global supply chain [15].

Understanding both domestic and international significance, various Vietnamese scholars have proposed legislative solutions, technological solutions, and sustainable business practices. Additionally, previous studies have suggested that customers' attitudes, cultural value, and long-term orientation (LTO) impacts their purchase intention toward green products [18, 19]. Since Vietnamese consumers follow a collectivistic culture [20], it is predicted that Vietnamese consumers are more likely to adopt green consumption. Vietnamese businesses are also familiar to the Triple Bottom Line (TBL) marketing, focusing on social and environmental issues when they do on profits: "Fashion and textiles produced by sustainable practices can alleviate the ecological and social strains, and provide an ethical choice for sustainable-conscious consumers to buy sustainable products"[16]. However, the sales of sustainable fashion firms did not show such welcoming adoption. Not many entrepreneurs have successfully survived and scaled up eco-friendly fashion firms. Leftover firms provide less attractive designs and styles than their fast fashion counterparts, which creates inconvenience in purchasing and an unfulfilled customer experience [21]. Therefore, the authors suspect that customers attitudes, knowledge, and behaviour and the current proposed solutions did not fully account for the interaction between the fashion industry and customers, leading to the low adoption of sustainable fashions in Vietnam.

Finally, as Vietnamese startups are still small with limited resources, funding for marketing strategies should be effectively utilized, concentrating on the most potential customer segmentation. In terms of eco-friendly fashion, generation Z (1996 to 2010) customers are particularly relevant because of their fast-growing willingness for sustainability and their value of supporting the environment [21, 22]. Members of this generation are educated customers who are well-versed in environmental issues and eco-friendly products. Consequently, they feel that businesses should be obligated to address and remedy environmental and social challenges [23]. Furthermore, they were willing to contribute by shifting their consumption to environmentally friendly products [24]. Therefore, they are potential early adopters of Vietnam's green fashion industry. In the near future, they will become adult parents responsible for purchasing clothing for their young children (succeeding generations such as Generation Alpha) and their old parents, such as Generation Y and X. Meanwhile, prior literature has examined the behavior of consumers toward green apparel in different countries such as Vietnam, China and the United States. However, a few studies emphasized generation Z as the study sample [25, 26]. This research also aims to enrich the empirical literature on young consumers in eco-

friendly fashion in developing countries. An important trait of sustainability is the consistent adoption of eco-friendly products rather than a purely trial. However, previous studies on Gen Z in Vietnam, India, Thailand, and Korea only have examined perceptions of purchase intention rather than customer loyalty [27, 28]. This study takes a further step to explore factors that influence customer loyalty—both quantitatively and qualitatively so that firms can benefit from context-rich data.

With the proposed global literature gaps, such as the lack of research on developing nations and Generation Z and the significant status of Vietnam in the global textile supply chain, the authors of this study chose Vietnam and Generation Z as research contexts and research subjects. We also employed a mixed-methods approach to analyze Vietnamese Generation Z customers' attitudes towards sustainable fashion, examining the following research questions:

RQ1: Which factors have a stronger impact on consumer loyalty and purchase intention toward eco-friendly fashion in Vietnam: developing quality eco-friendly products and services or promoting consumers' understanding of environmental preservation?

RQ2: From the interviewees' perspective, how to encourage the consumption of eco-friendly products in Vietnam?

## Literature review

Customer fulfillment is defined as the ability of companies or businesses to fulfill customer orders or requests with good interaction and customer support, on-time delivery, and after-sale services. This factor has been useful to companies and has greatly affected customer satisfaction since the early 1990s [29]. Therefore, customer fulfilment has received attention from companies in North America, Western Europe, and Japan. Previous studies in Taiwan on the relationship between after-sales services and customer satisfaction have demonstrated that service quality positively influences satisfaction, brand image, and brand loyalty in the fashion industry [30, 31]. However, little attention has been paid to how customer fulfillment affects customer satisfaction in Vietnam and to prior research on eco-friendly fashion.

Additionally, price value image is the customer's perceived value of a product in terms of quality and satisfaction regarding its price. In Taiwan and Fiji, two studies show that price value image significantly impacts customers' intention to purchase smartwatches and supermarkets' customer satisfaction, respectively [30, 32]. With regard to eco-friendly fashion products, PVI has been shown to affect customer satisfaction. One study in the context of the UK has shown patterns in the effects of price and qualities received on sustainable fashion use: while priced highly, sustainable fashion is perceived to have sufficiently higher quality, fewer health problems on the skin than other fashion, and as such, customers "see sustainable clothing as a net positive value alternative" [33]. As customers perceive eco-friendly fashion products to be premium in terms of both price and quality [25], the authors of this study believe that developing eco-friendly products would further increase the effect of price value image on customer satisfaction.

Customer satisfaction is the attainment of consumption patterns, showing the level of satisfaction with desires and expectations [34]. Customer satisfaction is a significant factor in the service sector in all industries because of its impact on business profits and performance improvement [35, 36]. Meanwhile, purchase intention refers to the likelihood of purchases made by customers in the future [37], and previous studies have pinpointed the link between customer satisfaction and purchase intention. One study also showed that when consumers are satisfied with a green product, their purchase intentions and tendencies to build brand loyalty increase in the context of South Korea, China, and Japan [38]. If the previous sustainable

purchasing experience fulfills their expectations and does not negatively affect their health or finances, they have more intention to purchase more sustainable fashion products. However, a recent study found little investigation into the relationship between customer satisfaction and purchase intention toward eco-friendly fashion products in developing countries [39]. To address the gap in customer fulfillment and examine the correlation between how customer fulfillment and price value image affect customer satisfaction in Vietnam, the following hypotheses are proposed:

H1: Customer fulfillment significantly impact customer satisfaction

H2: Price value image significantly impact customer satisfaction

H3: Customer satisfaction significantly impacts purchase intention

In addition to product and service quality, customer-related factors, such as environmental awareness, financial capacity, and willingness, also contribute to the consumption of eco-friendly fashion. The authors rely on the Theory of Planned Behavior (TPB) and the Theory of Reasoned Action (TRA) theory to investigate customer-related factors. While TRA explores customers' behavior based on customers' subjective norms and attitudes, The TPB model adds a new construct, "perceived behavioral control" (PBC), to represent customers' perception of ease or difficulty in doing the behavior of interest". TPB model is a response to the critics that TRA functions poorly for those with limited volitional control [25]. Researchers have employed TPB and TRA model because of its high predictability in pro-environment and green apparel research across Vietnam and globally [39, 40]. In addition to product and service quality, customer-related factors, such as environmental awareness, financial capacity, and willingness, also contribute to the consumption of eco-friendly fashion. Many studies found that in the initial phase of eco-friendly product purchase, consumers who value the environment can proceed to the stage of information seeking and searching for green items [12]. Marketers and businesses have relied on these findings to generate favorable attitudes from consumers toward firms' products through better communication and product image engagement [38, 41, 42].

However, empirical evidence of how consumer-driven factors, including perceived behavioral control and environmental concerns, impact purchase intention shows conflicting results. Many researchers have observed that consumers who are highly concerned about the environment and practice eco-friendly behavior are more likely to purchase green products [43]. In addition, through Twitter data mining, studies have revealed the factors that influence customers' purchase behavior for environmentally friendly products, demonstrating a strong relationship between customers' impression of eco-friendliness and the success of luxury fashion items [44, 45]. Thus, environmental concerns and PBC are often considered impetus for purchase and consumption.

On the other hand, in 1998, one paper studied consumers' apparel consumption practices related to the environment and found that environmental awareness and consciousness do not influence their purchase of apparel products [46]. However, this study shows that environmentally responsible behavior, a modified construct from PBC, significantly impacts environmentally responsible apparel consumption behavior. In contrast, two studies found that environmental concern significantly impacts the purchasing intention of young Bangladesh toward eco-friendly apparel, but PBC does not [45, 47]. Since there are different empirical results on the relationship between environmental concerns and PBC on purchase intention, culture and contexts might influence this relationship. Therefore, there is a need to assess this relationship in Vietnamese eco-fashion. Since the majority of the literature considers a significant positive relationship between these variables, the following hypotheses are proposed:

H4: Environmental concerns significantly impact purchase intention

H5: Perceived behavioral control significantly affects purchase intention

We need to understand the determinants of customers' purchase intention, satisfaction, and loyalties to further promote the conversion from non-sustainable fashion to sustainable fashion consumption. As customers' loyalties have re-enforcement abilities on one's behaviors and spread the usage to the networks, mastering the concept of sustainable fashion would directly increase the survival rates and growth of sustainable fashion businesses, making eco-friendly fashion products more available, more quality for society. On the social scale, this contribution would indirectly mitigate pollution issues related to unsustainable fashion.

Numerous previous studies have identified the influence of customer satisfaction and purchase intention on customer loyalty [48–50]. According to [51, 52], customer loyalty has been explained as a profound commitment to repurchase or frequently visit a preferred product or service in the future, regardless of the impact of marketing efforts on changing behaviors. Consumers with satisfying experiences from products tend to become loyal customers. Hence they continuously repeat their purchase behavior at that brand, which creates an opportunity for sustainable industry flourishment [53]. For instance, existing studies in the Korean traditional fashion market have found that customer satisfaction is a significant factor in developing a solid relationship between repurchase decisions and fashion products [51]. Investigating 1126 local participants from 14 fashion retailers in Bangladesh, one study indicated that a satisfied customer would become loyal [53]. Research conducted in the Hong Kong fashion retailing industry from 202 customers has also found that product satisfaction positively affects brand preference and repurchase decisions [54]. However, there are an insufficient number of studies on purchase intentions that impact loyalty to eco-friendly fashion products. Based on the above evidence, the following hypotheses are proposed:

H6: Customer satisfaction significantly impacts customer loyalty

H7: Purchase intention significantly impacts customer loyalty

According to one study in South Korea, China, and Japan, environmental concern drives the perceived benefits of consuming sustainable fashion, which leads to purchase and eWOM intention [38]. Another study conducted in Southern Gauteng, South Africa, with consumers who shopped at different shopping malls has shown that the consciousness of customers' environmental issues is positively related to their intentions to repurchase eco-friendly retail products [55]. Research has also stated that when customers find satisfactory eco-friendly products, there is a great likelihood that they will experience them again. Therefore, the following hypotheses were proposed:

H8: Environmental concerns for fashion production significantly impact customer loyalty.

H9: Perceived behavioral control significantly impacts customer loyalty.

With the qualitative findings and the literature gap, Fig 1 visualizes the conceptual model and research hypotheses:

## Methods

### Research design: Mixed-method approach

According to the literature, the mixed-methods approach permits researchers to investigate and observe insightful customer behaviors through in-depth interviews [56]. It also tested these observations with data-driven and rigorous analyses using a larger sample size. Therefore, to

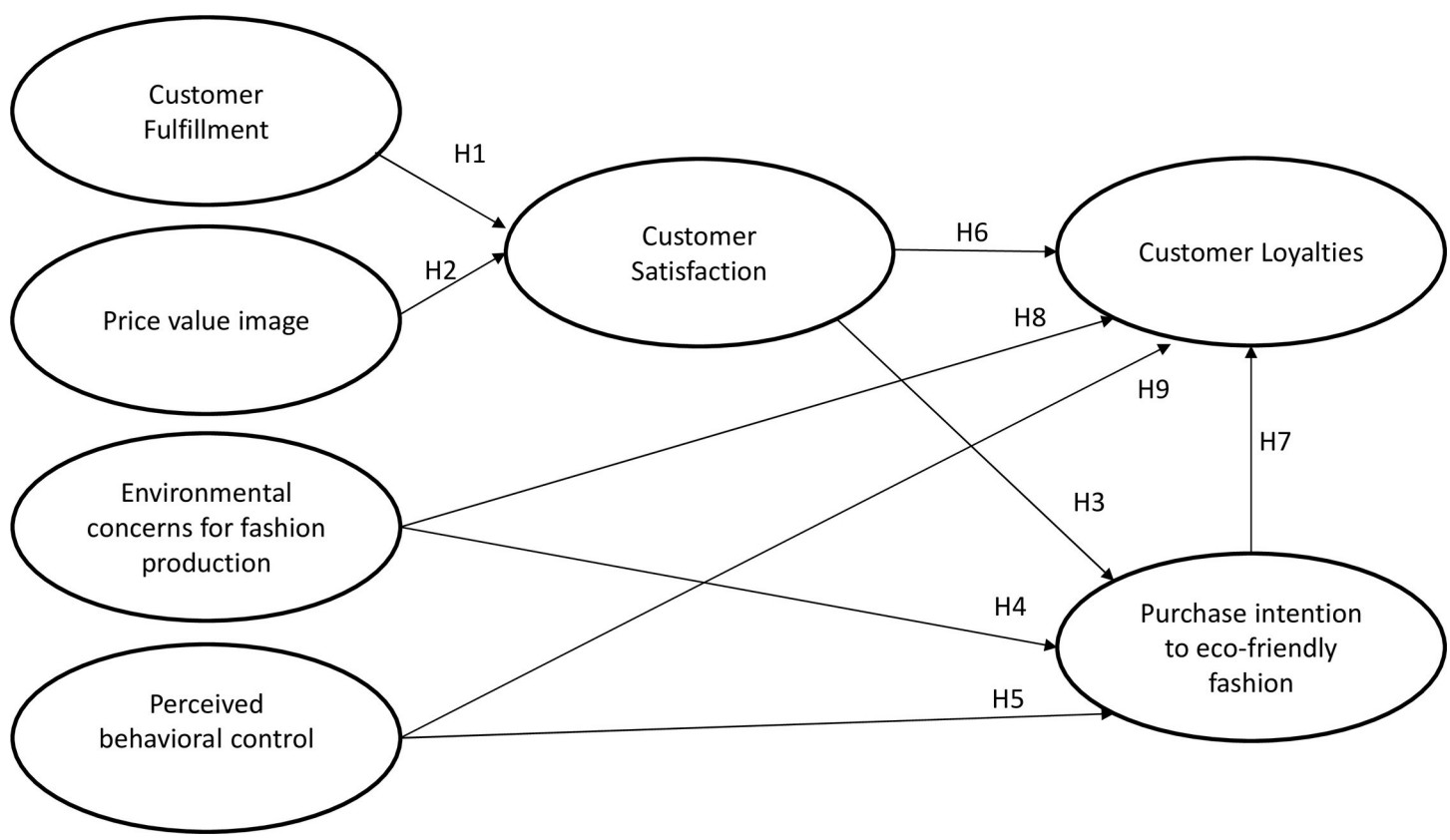

**Fig 1. The conceptual model and research hypotheses.**

understand Vietnamese Gen Z's attitudes towards sustainable fashion and confirm this conceptual model, the authors employed a mixed-methods approach and separated the research into two phases: qualitative and quantitative. While the study's qualitative results can generate data-rich suggestions from customers' perspectives to improve sustainable fashion, the quantitative results test the significance of the relationships and rank how each independent variable affects customers' purchase intention and customer loyalty. Social enterprises and fashion firms can use this data to navigate their decision-making processes and business strategies.

In the qualitative phase, the authors conducted in-depth and open-ended interviews and thematic analysis to understand the factors that influence Gen Z's intention to buy sustainable and green fashion products. The authors also followed the COREQ guidelines to depict the transparency and rigor of the data collection and analysis process [57]. The authors used the interview data from phase 1 to construct a conceptual model for the quantitative phase. In the quantitative phase, the authors designed and distributed a questionnaire to gen Z residents in Vietnam. After participants completed the questionnaire, the authors employed Partial Least Squares Structural Equation Modeling (PLS-SEM) to analyze the relationship between each construct in the model. This phase answers whether the constructs developed from the qualitative phase have a significant impact on customer loyalty and purchase intention in a sustainable fashion context.

## Phase 1: Qualitative study

Based on the study about thematic analysis, in-depth and semi-structured interviews allowed the authors to understand the thought process of participants regarding the decision to buy

sustainable fashion products [58]. More specifically, the structured questions aimed to investigate the demands, motivations, and challenges of participants when deciding between sustainable and non-sustainable fashion attire. On the other hand, unstructured sessions allow respondents to share their thoughts about the research topic which might be beyond the expectations of the authors, allowing the enrichment of information and diverse perspectives in this study. The authors further employed thematic analysis techniques to code and organize responses into valuable business suggestions and business constructs.

The interviewers in this phase were K.L, A.N, Y.T and Y.N The team had 1 male (K.L) and three female interviewers, all of whom had previous training in qualitative research and interview processes. Interviewers are currently research assistants in social science and business. They collected data from September 2021 and recruited a sample size of 24 participants via purposive sampling at high schools in Vietnam via social media (Facebook). Table 1 shows the participants' social demographics. Although the sample was small and unbalanced between male and female customers, this number of participants was suitable for the explorative nature of phenomenological and qualitative research [59]. Due to the Covid-19 pandemic, the interviewers met all the participants via Zoom, Google Meet or telephone and each session lasted from 7 minutes to 37 minutes. Moreover, before conducting the interviews, the interviewers sent out a Qualtrics form to collect the participants' demographic information, aiming to exclude any potential participants who were not born in the generation Z period (1996 to 2010) in accordance with the categorization of the Pew Research Center and younger than 16 years old to comply with the Children Law of Vietnam [22].

The interviews were conducted in 3 main sections. In the first section, the interviewers mentioned the research goals and clarified any questions about the technical concepts that might create difficulties for the participants. The participants were welcome to pose any questions about the research, their data privacy, and their rights regarding the study. At the end of

**Table 1. Demographic characteristics of participants in the qualitative phase.**

| Demographic Characteristics | n | % |
|---|---|---|
| **Generation Z** | 24 | 100 |
| **Gender** | | |
| Male | 6 | 25.0 |
| Female | 17 | 70.8 |
| Other/Not disclosed | 1 | 4.2 |
| **Education level** | | |
| Not graduated from junior high | 0 | 0 |
| Studying or graduated from high school | 19 | 79.2 |
| Studying or graduated from college | 5 | 20.8 |
| Studying or graduated from post-undergraduate education | 0 | 0 |
| **Living environment** | | |
| Urban | 14 | 58 |
| Suburban or Countryside | 10 | 42 |
| **Monthly family income approximation** | | |
| 0–7,499,999 VND | 4 | 16.7 |
| 7,500,000–14,499,999 VND | 6 | 25 |
| 15,000,000–29,999,999 VND | 6 | 25 |
| 30,000,000–44,999,999 VND | 4 | 16.7 |
| > 45,000,000 VNĐ | 4 | 16.7 |

*Exchange rate used: 1 USD = 22,760.50 VND

this section, the authors collect the written consent from the participants. The authors recorded all interview sessions with written notes and might also audio-tape the interviews whenever the participants were permitted. In the second section, the study follows the question guide (See S1 File), which is based on the pilot study, TBP, and knowledge-attitude-behaviour theory. In the third section, the interviewers and the participants talk more openly about sustainable fashion in a greater picture. This section allows interviewers to develop new constructs or amend the TPB theory construct to suit the context of sustainable fashion. Finally, the authors apply the member checking method with several participants to ensure that the study interpretation correctly reflects the thoughts of the participants, ensuring the credibility and validation of the analysis [60].

All authors performed data coding and analysis. While all authors have studied and performed data coding, TN and KT also have publications on qualitative studies. Before conducting the data coding, the authors translated the interview notes from Vietnamese to English using the forward-backward translation method. Finally, the authors followed the thematic analysis steps developed in [58]. The steps are summarized in Table 2:

## Phase 2: Quantitative study

The quantitative phase explores whether the relationships between variables in the conceptual model are statistically significant (Fig 1). The analysis also identifies the factors that have the strongest impact on the purchase intention of sustainable fashion products and their impact on customer loyalty. As the existing literature has mentioned the strength of the SEM-PLS technique in exploratory or an extension of an existing structural theory compared to CB-SEM, this research employs the SEM-PLS technique [61]. Second, this method is useful for the prediction of relationships and is more suitable for theoretical development than other methods, such as CB-SEM. This is important because the current research model is not heavily based on well-established theories. Third, this technique also allows analysis with a small sample size and has no assumptions regarding data distribution [61]. Finally, as customers' perceptions are very subjective and there are differences between cultures, this method is rigorous in analysis and easy to replicate in future research in another context.

The questions of the questionnaire were developed based on the theory of previous research while incorporating data from the qualitative stage to contextualize them in a sustainable fashion context. It collects data on 7 variables, environmental concerns for fashion production (Envf), perceived behavioral control (PBC), customer fulfillment (CF), price

**Table 2. 6 Steps to analyze the data via thematic analysis approach.**

| Thematic Analysis | Procedure for each part |
|---|---|
| **Step 1**: Familiarizing with the collected data | Familiarizing with the collected data by listening or writing repetitively about the written notes and audio. |
| **Step 2**: Creating initial codes | Coding recognizable patterns into keywords in a systematic approach. |
| **Step 3**: Constructing themes based on the codes | Assembling similar and closely related codes into umbrella themes. |
| **Step 4**: Reviewing and categorization themes into a coding tree | Organizing the coding tree by arranging the themes into suitably researched constructs. When new themes emerge, authors will discuss and re-group the themes. |
| **Step 5**: Naming and defining themes | Each theme will be named, defined, and assigned 1–2 quotations that can represent or provide evidence about the themes. |
| **Step 6**: Assembling the analysis into a final report | Analyzing the data by discussing the implication of the findings and compared with the previous literature or Vietnamese social norm. |

value image (PVI), purchase intention (PI), customer satisfaction (CS), and customer loyalty (CL). These seven variables have 28 items, and are measured on a seven-point Likert-type scale ranging from "strongly disagree" (1) to "strongly agree" (7) (See Table 4). Specifically, 4 items of the Envf construct are based on Gam, H.J.'s theory and our qualitative data [62]. 4 items on perceived behavioral control and 3 items on purchase intention were adapted from Kim studies [63]. The authors adapted 4 items on price value image from Zielke's study [64] and 5 items on customer satisfaction from Ndubisi [65], and 5 items on customer loyalty from Yee [66]. The authors used qualitative results to develop 4 items for customer fulfillment. The researchers also conducted a 30 minute focus group session with two fashion and environmentalist experts to ensure that the translations and adoptions of the items in the questionnaire were understandable, suitable and appropriate to the context of Vietnam and the industry and the validity of the measurement scale. Before distributing the questionnaire, a pilot study was conducted with nine participants to ensure readability and validity.

The authors used convenience and snowball sampling to recruit generation Z residents in Vietnam through a Qualtrics survey. These sampling techniques help researchers target the right participant group and achieve the sample size threshold in a short period of time (August to September 2021) with an affordable budget. We distributed the Qualtrics questionnaire on social media platforms, namely Facebook, on researchers' accounts in a public setting. We also posted on various Facebook groups in high school and university student communities and young professional communities to recruit more respondents. After cleaning the data, the study had a sample size of 313 participants, as summarized in Table 3. This sample size is larger than the recommended sample size for conducting the SEM-PLS method according to Soper study [67]. The authors recognized that the number of female participants was twice the number of male participants; hence, we used

**Table 3. Demographic characteristics of participants in quantitative phase.**

| Demographic Characteristics | n | % |
|---|---|---|
| **Generation Z** | 313 | 100 |
| **Gender** | | |
| Male | 87 | 27.8 |
| Female | 219 | 70.0 |
| Other/Not disclosed | 7 | 2.2 |
| **Education level** | | |
| Not graduated from junior high | 64 | 20.4 |
| Studying or graduated from high school | 185 | 59.1 |
| Studying or graduated from college | 64 | 20.5 |
| **Living environment** | | |
| Urban | 163 | 52.1 |
| Suburban or Countryside | 150 | 47.9 |
| **Monthly family income approximation** | | |
| 0–7,499,999 VND | 141 | 45.0 |
| 7,500,000–14,499,999 VND | 77 | 24.6 |
| 15,000,000–29,999,999 VND | 58 | 18.5 |
| 30,000,000–44,999,999 VND | 21 | 6.7 |
| > 45,000,000 VNĐ | 16 | 5.1 |

*Exchange rate used: 1 USD = 22,760.50 VND

**Table 4. Results of outer model measurement.**

| Code | Variables/Indicator | Loading | α | CR | AVE |
|------|---------------------|---------|---|----|----|
| **Perceived behavioral control** | | | 0.745 | 0.838 | 0.564 |
| PBC1 | I have the right to choose to purchase eco-friendly products rather than non-environmentally friendly products | 0.736 | | | |
| PBC2 | I am confident that I can purchase eco-friendly products instead of non-environmentally friendly products | 0.774 | | | |
| PBC3 | I can afford to purchase eco-friendly fashion products | 0.720 | | | |
| PBC4 | I have enough time to purchase eco-friendly fashion products | 0.773 | | | |
| **Consumer Fulfillment** | | | 0.858 | 0.903 | 0.701 |
| CF1 | When I shop at my favorite store, the customer care service is very friendly | 0.820 | | | |
| CF2 | My favorite shop takes good care of customers | 0.888 | | | |
| CF3 | My favorite shop has very good customer support policies | 0.848 | | | |
| CF4 | My favorite shop fulfills all the requirements that I need | 0.791 | | | |
| **Price Value Image** | | | 0.899 | 0.930 | 0.768 |
| PVI1 | In relation to store attributes, prices are very good in my favourite shop. | 0.880 | | | |
| PVI2 | In relation to product quality, prices are very good in my favourite shop. | 0.919 | | | |
| PVI3 | If I buy products in my favourite shop, I am confident I will get good quality for a good price. | 0.870 | | | |
| PVI4 | If I compare shop attributes to other shops, I think the prices are acceptable. | 0.835 | | | |
| **Environmental concerns for fashion production** | | | 0.717 | 0.835 | 0.629 |
| Envf1 | I am concerned about the negative impact of the fashion industry on the environment | 0.748 | | | |
| Envf2 | I believe that the fashion production process can be harmful to the environment | 0.766 | | | |
| Envf3 | I think It is important to develop eco-friendly fashion products | 0.860 | | | |
| **Purchase intention** | | | 0.817 | 0.891 | 0.732 |
| PI1 | I will prefer environmentally friendly products over non environmentally friendly products. | 0.831 | | | |
| PI2 | I am willing to purchase a product for environmental reasons. | 0.872 | | | |
| PI3 | I will make an effort to purchase products to protect the environment. | 0.863 | | | |
| **Customer Satisfaction** | | | 0.893 | 0.926 | 0.757 |
| CS1 | I am satisfied with the products and service provided. | 0.875 | | | |
| CS2 | My favourite shop has met all my expectations. | 0.845 | | | |
| CS3 | Compared with other shops, the level of satisfaction was high. | 0.861 | | | |
| CS4 | Overall, I am satisfied with this shop. | 0.899 | | | |
| **Customer Loyalty** | | | 0.868 | 0.905 | 0.655 |
| CL1 | I will do more transactions with my favourite shops in the coming years. | 0.791 | | | |
| CL2 | I will consider this shop as my first choice for purchases. | 0.821 | | | |
| CL3 | I will recommend this shop to friends who seek my advice on purchases. | 0.820 | | | |
| CL4 | I will say something good about this shop to others. | 0.787 | | | |
| CL5 | I will encourage friends and relatives to purchase from this shop. | 0.827 | | | |

MGA-PLS analysis to identify whether the genders of participants were significantly different regarding the relationship of constructs within the proposed model. The results showed no significant difference between male and female participants (Appendix 4).

The PLS analysis followed the procedure developed in Hair study [61]. First, we calculated the outer model using convergent validity, composite reliability, construct reliability, and discriminant validity. Regarding the inner model assessment, Table 4 shows the values for Cronbach's alpha (a), composite reliability (CR), and average variance extracted (AVE) and Table 5 shows the Heterotrait-Monotrait ratio. Second, the authors assessed the inner model, namely the testing of collinearity, R-Square ($R^2$) test, size effect test ($f^2$), relevant predictions ($Q^2$), and path coefficients. To analyze the data, the researcher employs the SmartPLS 3.3.3 to calculate the outer model and inner model.

**Table 5. Heterotrait-monotrait ratio.**

|      | CF    | CL    | CS    | Envf  | PBC   | PI    | PVI  |
|------|-------|-------|-------|-------|-------|-------|------|
| CF   |       |       |       |       |       |       |      |
| CL   | 0.655 |       |       |       |       |       |      |
| CS   | 0.749 | 0.792 |       |       |       |       |      |
| Envf | 0.327 | 0.350 | 0.374 |       |       |       |      |
| PBC  | 0.488 | 0.614 | 0.594 | 0.519 |       |       |      |
| PI   | 0.604 | 0.645 | 0.599 | 0.484 | 0.822 |       |      |
| PVI  | 0.697 | 0.616 | 0.792 | 0.350 | 0.559 | 0.486 |      |

## Results

### Qualitative results

After interviewing 24 participants, the qualitative phase generated content-rich data about the interviewees' level of environmental awareness. Most interviewees reported being aware of eco-friendly fashion products. The primary sources of information about eco-friendly products are social media. Participants mentioned that they knew about eco-friendly products through posts and videos from influencers, environmental projects, and organizations. Some said they read fashion e-newspapers and magazines and gained insight into the term eco-friendly products. In their definition, eco-friendly fashion products are clothes, jewelry, and accessories made from eco-friendly materials and manufactured in an eco-friendly process such as tote bags and tire sandals.

Despite their environmental awareness, only six interviewees used eco-friendly products, while the rest did not. One of the primary reasons for not using this type of product is that they do not have sufficient knowledge to differentiate between normal and eco-friendly products. Because eco-friendly fashion shops do not run marketing campaigns and reveal that these products are eco-friendly when potential customers visit the shop, they do not know this property to buy. In addition, they do not consider environmental factors when purchasing. More than half of the participants said that when they buy clothing, they focus mainly on the quality and design of the product. The materials, diversity of styles, suitability and fit of the product should match their interests and expectations. Since they believe that eco-friendly fashion brands have limited options, they do not intend to visit these shops. In addition, most interviewees reported that they were students, and hence a small budget for shopping. They found that eco-friendly products usually have higher prices than their normal counterparts. Even people who use eco-friendly fashion products consider the costs before other factors when they purchase. Finally, eight interviewees said that the unavailability of eco-friendly products prevented them from consuming. Although they intend to purchase eco-friendly products, they cannot find the shop to buy. Besides, as eco-friendly products are not popular in their areas, finding eco-friendly products is challenging for potential customers.

In terms of factors motivating them to consume eco-friendly products, price, product performance, and service quality mainly determine interviewees' purchase and repurchase intentions. Sixteen interviewees believe that if their price is reasonable, the same, or just a little higher than normal products, they will consider these products. However, prices also need to satisfy their expectations of product quality. As fabric material, content and stitches will significantly affect the product's visual, 15 participants suggested that if the product fails to make them feel comfortable, they will not buy them regardless of price. Some interviewees also care about the durability of the products; hence, the patterns and color should maintain the same after a long time of use. Considering patterns, colors, and design style as a tool to boost

confidence in front of the public, participants believe that they will prefer eco-friendly products if they deliver diverse options as normal products. Some interviewees were willing to purchase and recommend their friends to buy eco-friendly products with higher prices if they had good quality because they perceived environmental factors as added value. Besides, six interviewees believe that they will repurchase eco-friendly products if they can confirm that the shop does not fake the product's eco-friendliness. The level of environmental awareness of salespersons and shop owners illustrated in their first visit also affects their repurchase. This is because they can ensure the honesty of the shop, ensuring the eco-friendliness characteristics of products. Service quality also plays a crucial role in repurchase intentions. Young people will revisit the shops if they feel politely welcomed by salespersons at the time of their first purchase. They also feel encouraged if the shops offer sale-off programs and discounts for loyal customers. Overall, perceived product quality and service quality are the two driving forces behind customers' purchase behavior. The following hypotheses are posited:

H10: Perceived behavioral control more significantly impacts purchase intention than customer satisfaction and environmental concerns regarding eco-friendly fashion products.

H11: Customer satisfaction has a greater impact on customer loyalty than perceived behavioral control and environmental concerns regarding eco-friendly fashion products.

Regardless, to encourage people to switch to more eco-friendly alternatives, interviewees recommended that two optimal strategies are educating people about environmental effects and promoting eco-friendly fashion products. First, the interviewees perceived that young people were aware of eco-friendly products. However, they do not comprehensively understand the positive impacts of these products on the environment or the current situation of the unsustainable world of fashion. Therefore, instilling more knowledge and information into people's minds can automatically turn into environmentally conscious actions. Second, promoting eco-friendly fashion products via social media and influencers' endorsements will help potential customers remember this product when people are environmentally conscious. One of the main problems is that people do not know places to buy eco-friendly products; thus, the brands should also conduct marketing campaigns to help people feel easy to buy. In this way, customers will feel an increased availability and accessibility of the products.

## Quantitative results

**Outer model assessment.** The Outer Model Assessment of reflective constructs addresses the indicator reliability, internal consistency reliability, convergent validity, and discriminant validity of the empirical model in the marketing context. All thresholds in this study followed the guidelines developed in [68]. After excluding 2 indicators (ENVF4, PBC1) because their loading factor was lower than 0.70. Table 4 shows the assessment of loading factor, Cronbach's alpha, composite reliability, and average variance extracted from six constructs: customer fulfillment, customer loyalty, customer satisfaction, environmental concerns for fashion production, perceived behavioral control, purchase intention, and price value image. According to this table, all constructs have an AVE score from 0.564 to 0.768 ($> 0.5$), Cronbach's alpha score from 0.717 to 0.899 ($> 0.7$), and composite reliability score from 0.835 to 0.930 ($> 0.7$), satisfying the measurement benchmarks [68]. Regarding the test for discriminant validity, the Heterotrait-monotrait ratio (Table 5) showed that all values were higher than 0.90, satisfying the benchmark for discriminant validity [69]. As all measurements meet all benchmarks in the Outer Model Assessment established in the literature, the research moves to the Inner model Assessment.

**Inner model assessment.** The Inner Model Assessment examines the Collinearity test, the R-square ($R^2$), The size effect test ($f^2$), the Relevant prediction test ($Q^2$), and the Path analysis of the empirical model [68]. First, the study examines Collinearity Issues to ensure that there is no bias in the regression results. According to the PLS results in Table 6, the structural model Collinearity (VIF) has coefficient values ranging from 1,138 to 1,947. As all coefficient values are smaller than the ideal threshold of 3.0, as suggested by Hair, the data show that there are no problems of collinearity [68].

Regarding the test for explanatory power, one study remarks that $R^2$ values from 0.50 to 0.75 show that the endogenous latent variable has moderate predictive power to the model and an R-value smaller than 0.50 shows a weak predictive power to the model [68]. Regarding the R-square analysis in this study, customer satisfaction, customer loyalty, and purchase intention have $R^2$ values of 0.587, 0.542, and 0.486 respectively. This result indicates that while price value image and customer fulfillment can explain 58.7% of customer satisfaction variants, environmental concerns for fashion production, customer satisfaction, purchase intention, and perceived behavioral control can explain 54.2% of customer loyalty variants. These two explanations have a moderate predictive power. On the other hand, customer satisfaction, perceived behavioral control, and environmental concerns for fashion production can only explain 48.6% of purchase intention variants with weak predictive power. In addition to the $R^2$ test, Hair study states that the relevant prediction test ($Q^2$) also shows the predictive value of the model if $Q^2 > 0$ [68]. The $Q^2$ coefficient value > 0,25 depicts a medium predictive accuracy for the PLS path model, and all constructs in this study meet this threshold. More specifically, customer satisfaction has a $Q^2$ value of 0.438 and both purchase intention and customer loyalty have a $Q^2$ value of 0.346. The $R^2$ and $Q^2$ results show that the proposed model of this study has moderate predictive values regarding how service quality, product quality, customer satisfaction, customer's environmental awareness, and perceived behavioral control on purchasing intention and customer loyalty toward sustainable fashion in Vietnam.

After processing the data using 5000 bootstrapped samples, the results support seven hypotheses and reject nine hypotheses at 95% confidence (Table 6). First, the data show that price value image (0.489, P = 0.000) and customer fulfillment (0.361, P = 0.000) show a significant relationship with customer satisfaction; hence, hypotheses 1 and 2 are accepted. Second, customer satisfaction (0.260, P = 0.000), environmental concerns for fashion production (0.184, P = 0.000), and perceived behavioral control (0.456, P = 0.000) significantly affected purchase intention; therefore, the study accepts hypotheses 3, 4, and 5. Finally, while customer satisfaction (0.553, P = 0.000) and purchase intention (0.199, P = 0.000) significantly influenced customer loyalty, the results supported hypotheses 6 and 9. However, environmental concerns for fashion production (0.019, P = 0.630) and perceived behavioral control (0.089,

**Table 6. Hypothesis testing results.**

| Path | Hypothesis | B | P-Values | $R^2$ | $f^2$ | $Q^2$ | IVF | Result |
|---|---|---|---|---|---|---|---|---|
| CF -> CS | H1 | 0.361 | 0.000 | 0.587 | 0.196 | 0.438 | 1,607 | Supported |
| PVI -> CS | H2 | 0.489 | 0.000 | 0.587 | 0.361 | 0.438 | 1,607 | Supported |
| CS -> PI | H3 | 0.260 | 0.000 | 0.486 | 0.102 | 0.346 | 1,291 | Supported |
| Envf -> PI | H4 | 0.184 | 0.619 | 0.486 | 0.058 | 0.346 | 1,138 | Supported |
| PBC -> PI | H5 | 0.456 | 0.000 | 0.486 | 0.322 | 0.346 | 1,257 | Supported |
| CS -> CL | H6 | 0.553 | 0.079 | 0.542 | 0.469 | 0.346 | 1,423 | Supported |
| Envf -> CL | H7 | 0.019 | 0.619 | 0.542 | 0.001 | 0.346 | 1,204 | Not Supported |
| PBC -> CL | H8 | 0.089 | 0.079 | 0.542 | 0.010 | 0.346 | 1,661 | Not Supported |
| PI -> CL | H9 | 0.199 | 0.000 | 0.542 | 0.045 | 0.346 | 1,947 | Supported |

P = 0.087) did not show a significant relationship with customer loyalty. Therefore, this study rejects hypotheses 7 and 8.

Regarding the hypotheses developed after the qualitative phrase, the $f^2$ test can compare the substantive impacts of each predictor construct on a dependent construct. According to two key studies [68, 70], thresholds for the $f^2$ test, $f^2$ effect sizes larger than 0.02, 0.15, and 0.35 show small, medium, and large impacts on dependent constructs respectively. The results confirm that perceived behavioral control has a higher impact on purchase intention than customer fulfillment and environmental concerns for fashion production, accepting hypotheses 10. More specifically, only perceived behavioral control has medium-size effects (0.35 > 0.332 < 0.15) on purchase intention while customer satisfaction (0.102 < 0.15) and environmental concerns for fashion production (0.058 < 0.15) only have a small effect. Finally, only customer satisfaction shows a large effect (0.469 > 0.35) on customer loyalty while purchase intention (0.045 < 0.15) only shows a small size effect. Perceived behavioral control (0.010 < 0.02) and environmental concerns for fashion production (0.001 < 0.02) did not significantly affect customer loyalty. This finding aligns with hypothesis 11 that customer satisfaction has a larger impact on customer loyalty than purchase intention, perceived behavioral control, and environmental concerns for fashion production.

## Discussions

In a recent study on green apparel, its limitation session questioned whether environmental concern and customer satisfaction can influence the purchase intention of customers in developing countries [39]. This study confirms that customer satisfaction, environmental concern, and PCB have statistically significant impacts on the purchase intention of the Vietnamese generation Z concerning eco-friendly fashion, addressing the contextual gap. This study also confirms the conflicting nature of customer-driven factors (environmental concerns and PBC) in purchasing intention. As both environmental concerns and PBC significantly influence the purchase intention of eco-friendly fashion in Vietnam, this study only partially supports previous studies of [47, 52]. In addition, while recent studies only studied the factors that impact purchase intention of eco-friendly fashion in the younger generation [27, 45], our study takes further steps to explore which factors influence customer loyalty and compares the importance of each factor on purchase intention and customer loyalty with the f square test. This mixed-method result and the comparison of effect sizes provide important theoretical and practical implications for sustainable eco-fashion.

Quantitative analysis shows that while PBC has a moderate effect on purchase intention; environmental concern and customer satisfaction have only a small effect. This section sheds light on this relationship. First, the Vietnamese generation Z may have knowledge and attitude about the environment, but do not have enough resources or motivation to buy eco-friendly products. The main barrier faced by most respondents in the qualitative phase was the lack of available eco-friendly products; hence, only customers who truly had spare time and financial resources were willing to buy these products. Many interviewees expressed that their lack of eco-friendly fashion consumption can be traced back to their lack of access to local eco-friendly shops and brands. In addition, customer-driven factors, namely environmental concerns, and PBC do not significantly affect loyalty. This relationship is evident in both qualitative and quantitative phrases. These insignificant results contradict the findings in previous studies. This qualitative finding also implies that Vietnamese youth do not automatically associate eco-friendly fashions with high-quality products, which is contrary to the findings in previous study [21].

Moreover, the international and Vietnam-based studies from producers' perspectives identify five major elements influencing customers' purchase intention of eco-friendly products:

transparency, influencers, acculturation, and self-esteem [71, 72]. Our study found only 3 of these elements, and no Gen Z participants mentioned acculturation. The difference between customers' perceptions in this study and producers' perceptions in previous studies highlights the gap between these stakeholders. Moreover, many previous studies have assumed that Vietnamese consumers have collectivist cultures. However, at least in our findings on the eco-friendly fashion industry, Generation Z participants illustrated individualism in both quantitative and qualitative results. Our participants concentrated on product quality (PIV), self-esteem, and resources (CPB) in making decisions rather than their knowledge of environmental consequences (Enfv). This new finding might be explained by the growth of Vietnamese Generation Z in a peaceful and modernized country without worrying about survivability—the pressure that previous generations must collectively join hands to achieve. Their values might shift from collectivism to individualism due to Western influences from globalization and more commodity [73]. With this contrasting finding from the literature, businesses and scholars should facilitate more research on generation Z to capture their unique nature, preventing stereotyping and outdated data from hampering effective business and marketing decisions.

Notably in the qualitative section, most interviewees suggested that firms should raise awareness and increase the accessibility of eco-friendly products to promote sustainable consumption. However, the SEM-PLS results illustrate that Gen Z's knowledge and awareness are insignificant compared to perceived product quality and control behaviors. Hence, the contradiction of Gen Z's suggestions in the qualitative phase and their responses in the quantitative phase implies that Gen Z's perceptions might not be consistent with their actions. This contradiction may also imply that the qualitative phase is vulnerable to the social desirability bias. Future research should investigate their behavior from more rigorous psychological methods.

In terms of managerial implications, these findings and comparisons have practical implications at both micro and macro scales. The mixed-methods approach allows fashion manufacturers and retailers to develop both data-driven and design-thinking and customer-centric strategies. On micro scale, Rajkishore Nayak mentioned in his recent study that there is a lack of research on sustainable development in fashion firms in Vietnam [72, 74]. He identified factors the that prevent Vietnamese fashion firms from adopting eco-friendly production by interviewing suppliers. Financial constraints, lack of access to modern technologies, lack of understanding, and difficulty finding potential markets for sustainable products are challenges for SMEs. From understanding the problems faced by Vietnamese eco-fashion firms in budgeting and market identification, our study concentrates on customers' perspectives to identify their needs and criteria for evaluating eco-friendly fashion products. Our study shows that Vietnamese firms should pay more attention to their services and products in the case of financial and resource constraints rather than investing in green marketing to educate Gen Z customers. This is driven by the data that product-related quality has a stronger influence on customers' behavioral intentions than environmental awareness. Moreover, since the PBC plays the most important role in stimulating purchase intention among young customers, eco-friendly fashion businesses should target upper-income groups of young customers. On the macro scale, the government should prioritize financial and technological support for fashion firms to develop high-quality eco-friendly fashion and optimize their logistics to ensure the availability of products in the market, ensuring the availability of the products to the majority of Gen Z customers. Some industrial strategies and notes on these policies have been developed from consultant firms, such as the McKinsey and Boston Consultant Group, and the World Economic Forum [75, 76].

In terms of theoretical implications, the research also highlights and compares the intensity of the effect of these factors on customer loyalty—a variable that has been underlooked in the

literature on eco-friendly fashion in Vietnam. The study also suggests that future research should investigate how social media promotes eco-friendly fashion within the youth generation. Second, this study addresses the research gap and conflicts mentioned in the literature on the roles of environmental concern and PBC on purchase intention in Vietnam and the eco-friendly fashion context. The inconsistency between this study and other empirical papers in different contexts shows that context may play an important role in young behavior intention toward eco-friendliness. As the result may not be transferable globally, researchers across the world should conduct research on their particular contexts. Finally, rather than investigating the relationship between these variables purely with numerical data as in previous studies, the data-driven results are also aided with content-rich and insight from in-depth interviews with participants to provide a potential explanation/ justification for the customer's behavioral intention. The qualitative section shows the potential role of social media in stimulating the purchase intention and awareness of young customers in an eco-friendly fashion. Therefore, shop owners in Vietnam can also attempt to adopt this media channel to acquire new customers and increase the presence of their products. However, this relationship has conflicting views on the empirical evidence. For instance, previous studies only confirmed the importance of social media on purchase intention in the cosmetic industry but not in the organic clothing industry [77, 78]. Hence, future research can test this relationship to investigate whether social media can have a strong influence on purchase intention in eco-friendly clothing in Vietnam.

This study has some limitations. First, the study only concentrates on generation Z in Vietnam; hence, the data cannot be generalized to different contexts and age groups. Future research should focus on other age groups and countries. Second, the research only investigates the factors, motivations, and barriers that influence the purchase intention and customer loyalty of Vietnamese youth to suggest potential policies and management strategies. Future research should employ a cost-benefit analysis or pre-post analysis to measure and compare the effectiveness of potential solutions. Finally, this study only investigated three service-product quality variables (CF, PVI, and CS) and two customers' perception and behavior variables (PCB and ENVF). These variables can only explain 48.6% of purchase intention variants and 54.2% of customer loyalty variants with weak and moderate predictive power respectively. Future research should further investigate other aspects that might influence the purchase and loyalty of customers, such as the role of technology, social influence, green innovation, and customer engagement with the firm. These findings can help firms and governments develop a more comprehensive strategy to promote the widespread and sustainable adoption of eco-friendly fashion products, which in turn benefits society in the long run.

## Conclusion

This study employs a mixed-methods approach to study how customers' perceptions of product-service quality and their environmental awareness and capacity impact the purchase intention and customer loyalty toward eco-friendly fashion products among Vietnamese Gen Z. The qualitative results further show that the knowledge and attitude toward eco-friendly fashion practices are not sufficient to convince young customers to afford eco-friendly fashion products. The quantitative results show that while customer-related factors play a more significant role in stimulating purchase intention, only product-service perceived quality factors impact loyalty. From Generation Z interviewee' perspectives, to promote the adoption of eco-friendly fashion in Vietnam, on micro scale, eco-friendly fashion shop owners in Vietnam should pay more attention to their services and products rather than investing in green marketing to educate customers on environmental awareness because product-related features has stronger influence on customers' behavioral intention. On the macro scale, the government

should prioritize financial and technological support for fashion firms to develop high-quality eco-friendly fashion and optimize their logistics to ensure the availability of products in the market, ensuring the availability of the products to the majority of Gen Z customers.

## Supporting information

**S1 File. Question guide for semi-structured interview.**
(PDF)

**S2 File.**
(PDF)

## Acknowledgments

The authors would like to thank all participants and academic experts because their opinions and contributions help us formulate the research presentation.

## Author Contributions

**Conceptualization:** Khoa Tran, Tuyet Nguyen, Yen Tran.

**Data curation:** Khoa Tran, Tuyet Nguyen, Yen Tran, Anh Nguyen, Khang Luu, Y. Nguyen.

**Formal analysis:** Khoa Tran, Tuyet Nguyen, Yen Tran, Anh Nguyen, Khang Luu, Y. Nguyen.

**Investigation:** Yen Tran, Anh Nguyen, Khang Luu, Y. Nguyen.

**Methodology:** Khoa Tran.

**Project administration:** Khoa Tran, Tuyet Nguyen.

**Supervision:** Khoa Tran.

**Validation:** Khoa Tran, Tuyet Nguyen, Yen Tran, Anh Nguyen, Khang Luu, Y. Nguyen.

**Visualization:** Khoa Tran, Tuyet Nguyen, Yen Tran, Anh Nguyen, Khang Luu, Y. Nguyen.

**Writing – original draft:** Khoa Tran, Tuyet Nguyen, Yen Tran, Anh Nguyen, Khang Luu, Y. Nguyen.

**Writing – review & editing:** Khoa Tran, Tuyet Nguyen.

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
