## [Decision Letter · Decision Letter 0]

28 Feb 2022

PONE-D-21-36570Promoting Eco-friendly Fashion among Youth Generation in Developing Countries: Mixed-methods study on The Roles of Price Value Image, Customer Fulfillment, and Pro-environmental BehaviorPLOS ONE

Dear Dr. Nguyen,

Thank you for submitting your manuscript to PLOS ONE. After careful consideration, we feel that it has merit but does not fully meet PLOS ONE’s publication criteria as it currently stands. Therefore, we invite you to submit a revised version of the manuscript that addresses the points raised during the review process. Both reviewers pointed out weaknesses in the paper. Please refer to their reviews for details. 

We look forward to receiving your revised manuscript.

Kind regards,

Maurizio Naldi

Academic Editor

PLOS ONE

Journal Requirements:

2. Please include a complete copy of PLOS’ questionnaire on inclusiveness in global research in your revised manuscript. Our policy for research in this area aims to improve transparency in the reporting of research performed outside of researchers’ own country or community. The policy applies to researchers who have traveled to a different country to conduct research, research with Indigenous populations or their lands, and research on cultural artifacts. The questionnaire can also be requested at the journal’s discretion for any other submissions, even if these conditions are not met.  Please find more information on the policy and a link to download a blank copy of the questionnaire here: https://journals.plos.org/plosone/s/best-practices-in-research-reporting. Please upload a completed version of your questionnaire as Supporting Information when you resubmit your manuscript.

Reviewers' comments:

Reviewer's Responses to Questions

**Comments to the Author**

1. Is the manuscript technically sound, and do the data support the conclusions?

Reviewer #1: Yes

Reviewer #2: No

2. Has the statistical analysis been performed appropriately and rigorously? 

Reviewer #1: Yes

Reviewer #2: Yes

3. Have the authors made all data underlying the findings in their manuscript fully available?

Reviewer #1: No

Reviewer #2: No

4. Is the manuscript presented in an intelligible fashion and written in standard English?

Reviewer #1: Yes

Reviewer #2: Yes

5. Review Comments to the Author

Reviewer #1: First of all, I am glad to have the opportunity to read the article on “Promoting Eco-friendly Fashion among Youth Generation in Developing Countries: Mixed-methods study on The Roles of Price Value Image, Customer Fulfillment, and Pro-environmental Behavior”, that I have read with great interest.

Undoubtedly, the subject matter addressed in this work is of considerable interest. However, from my humble point of view, the paper has great weaknesses.

In spite of these problems, and considering the interest of the subject, I would like to suggest that the authors revise their paper considering the comments offered below. In my view, the revised version should undergo a new assessment process.

Now, I would like to make some comments and suggestions that should always be understood in a positive way and considering that the different observations constitute different avenues that may allow improving this interesting research and facilitate its publication and impact in the subsequent specialized literature. With this initial caveat in mind, I would like to make the following observations and recommendations to the authors for their reflection and introduction of the changes they consider appropriate:

I would like to say that the topic of this paper is relevant, but from my humble point of view, the paper should be improved in the following points:

1) The title is very long. I recommend that the number of words be reduced.

2) The research methodology should be reflected in the abstract.

2) In the INTODUCTION section, the authors could explain what the research questions are.

3) In the LITERAQTURE REVIEW, the authors could include a better explanation of the loyalty and fashion sector. In this sense, the authors could include some relevant and references from the fashion sector such as:

Vinhas Da Silva, R. and Faridah Syed Alwi, S. (2006), "Cognitive, affective attributes and conative, behavioural responses in retail corporate branding", Journal of Product & Brand Management, Vol. 15 No. 5, pp. 293-305. https://doi.org/10.1108/10610420610685703

Kuikka, A. and Laukkanen, T. (2012), "Brand loyalty and the role of hedonic value", Journal of Product & Brand Management, Vol. 21 No. 7, pp. 529-537. https://doi.org/10.1108/10610421211276277

Cuesta-Valiño, P., Gutiérrez-Rodríguez, P. and Núnez-Barriopedro, E. (2021), "The role of consumer happiness in brand loyalty: a model of the satisfaction and brand image in fashion", Corporate Governance, Vol. ahead-of-print No. ahead-of-print. https://doi.org/10.1108/CG-03-2021-0099

4) In the METHOS section, the authors could explain better all the steps carried out in the process of obtaining the data for this investigation.

Why have only 24 people been selected? For example, why are there 6 male and 17 female?

Is the selected sample (313) representative and enough? Are these people representative of the population the authors want to study? Why are there so many more women than men?

The authors should explain better why they have used the SEM-PLS for the investigation.

5) The DISCUSSIONS section is very poor. Therefore, the authors should improve much this section on theoretical implications and practical implications, and they should end the section with the limitations of their study and the most important conclusion of their research.

For all these reasons, I think this article needs a minor revision, but I hope the authors will be able to do it correctly.

Reviewer #2: Thank you for the opportunity to review this paper. The research was very interesting particularly through the cultural lens of Vietnam. The introduction section successfully introduces the background context to which the research resides. Whilst this section was well written and the writing style has a nice flow I would have liked to have been given a sense of purpose at the beginning of the introduction section. This would have included locating the study in Vietnam.

The introduction covers policy and legislation from international governments, businesses again with an international approach, corporate social responsibility, and consumer behaviour. Collectively, this leads to a lack of clarity Regarding which literature stream this paper contributes to. Direction of the paper should be clearer by this stage.

It would be good to set the cultural context of Vietnam, there are discussions around behaviours and markets in global contexts and it would be good to know how this directs the research in Vietnam and the implications this has for the findings. I would have liked to have known what the fashion retail environment within Vietnam was like, for example where there are many international brands or Vietnamese brands are there large chain stores or are there smaller boutiques a better understanding of this would help to understand the whole context of the paper.

Additionally, I wonder what the link is between the variables in Figure 1 and sustainability, this could be clearer.

To address generation z Vietnam consumers opinions on sustainability and the fashion industry is highly relevant and makes a valid contribution to the literature. However, this could be more strongly linked to the idea about post purchase satisfaction and the other variables mentioned.

Focused research questions at the end of the introduction would help to better understand what the research seeks to investigate.

This sense of purpose would also help positioning of the literature review and this could be further established within the development of the literature reviewed. Clearer clarification on what the research is investigating would be helpful. The literature review begins also by covering a number of topics and would benefit from focusing more acutely on the research agenda.

Links between sustainable fashion and post consumption satisfaction from other contexts could be more clearly articulated within the literature review.

Often times references were missing to support points made and knowledge development. Additionally, a lot of research around sustainable fashion consumption behaviours, particularly, using the TRA was not included within the literature discussion. deepening this discussion would have helped create a stronger theoretical construct. I would have liked to have seen a more developed literature review that focused specifically on sustainable fashion consumption behaviours and what is already known about them.

p. 6 According to [24-26] - should this have the authors names? I found this a few times. Such as on page 11: According to [54],

I especially like to the table in which the data analysis steps were presented undescribed. The methodology section was well explained offering transparency on how data were collected. I would have liked to have known how the quantitative sample were accessed and communicated with.

The qualitative data was presented in quantitative manner and this did not communicate the perceived rich data that is typical of a qualitative study. There were no participant quotes presented in support of analytic development. I liked that the qualitative data was used to further to develop hypothesis and this helps to tighten up the methodological approach.

The data from the qualitative study is highly similar to the consumer behaviour sustainable fashion literature that has existed for the last 20 years. It would be good to have seen some kind of advancement on previous knowledge and perhaps it could have come from a cultural aspect.

Overall I would have liked to see the strong consistency within this paper setting out clear parameters that would be under investigation and link to the conclusions and discussion.

6. PLOS authors have the option to publish the peer review history of their article (what does this mean?). If published, this will include your full peer review and any attached files.

Reviewer #1: No

Reviewer #2: No

---

## [Author Response · Author response to Decision Letter 0]

28 Jun 2022

Dear PLOS One Editorial Office

Editor-in-Chief and Reviewers

The authors express our gratitude to the editors and reviewers for your advice on the manuscript. Your expertise has helped us improve the paper's organization and ensure our data collection and analysis are more transparent to the readers. After addressing your suggestions, we see a significant improvement in the clarity and quality of our study. We have addressed your comments in the subsequent sections and hope that our changes can meet your expectations. In our responses, your comments are written in red and our responses are written in black. The yellow highlights are excerpts from our new manuscript version.

In this version, we also requested English editing support from two native writers to enhance the clarity and coherence of the manuscript. We also attached the language reports from Pubsure and American Journal Experts (AEJ)’s grammar checking results.

Above all, the authors wish all editors and reviewers a fruitful academic year. Once again, thank you for your advice on this manuscript.

Best regards,

The authors

Reviewer #1: First of all, I am glad to have the opportunity to read the article on “Promoting Eco-friendly Fashion among Youth Generation in Developing Countries: Mixed-methods study on The Roles of Price Value Image, Customer Fulfillment, and Pro-environmental Behavior”, that I have read with great interest.

Undoubtedly, the subject matter addressed in this work is of considerable interest. However, from my humble point of view, the paper has great weaknesses.

In spite of these problems, and considering the interest of the subject, I would like to suggest that the authors revise their paper considering the comments offered below. In my view, the revised version should undergo a new assessment process.

Now, I would like to make some comments and suggestions that should always be understood in a positive way and considering that the different observations constitute different avenues that may allow improving this interesting research and facilitate its publication and impact in the subsequent specialized literature. With this initial caveat in mind, I would like to make the following observations and recommendations to the authors for their reflection and introduction of the changes they consider appropriate:

Dear Reviewer 1,

Thank you very much for your comments. It is very constructive and we have made the necessary changes according to your suggestions. We hope to hear more comments from you and thank you for your support. 

I would like to say that the topic of this paper is relevant, but from my humble point of view, the paper should be improved in the following points:

1) The title is very long. I recommend that the number of words be reduced.

The current version has 16 words, which is a normal length for an article. We have shortened our title into

Eco-friendly Fashion among Generation Z: Mixed-methods study on Price Value Image, Customer Fulfillment, and Pro-environmental Behavior.

2) The research methodology should be reflected in the abstract.

We have updated the following information:

Thus, this study employs a mixed-methods approach with thematic analysis and the SEM-PLS technique to research how Vietnamese Gen Z's perceptions of product-service quality, environmental awareness, and pro-environmental behavior influence their purchase intention and loyalty toward eco-friendly fashion products.

2) In the INTRODUCTION section, the authors could explain what the research questions are.

We have added the following paragraph:

We also employed a mixed-methods approach to analyze Vietnamese Generation Z customers' attitudes towards sustainable fashion, examining the following research questions:

 RQ1: Which factors have a stronger impact on consumer loyalty and purchase intention toward eco-friendly fashion in Vietnam: developing quality eco-friendly products and services or promoting consumers' understanding of environmental preservation?

 RQ2: From the interviewees’ perspective, how to encourage the consumption of eco-friendly products in Vietnam?

In the newest version of the Introduction, we have truncated the non-management theories and information you suggested. We also re-organized our introduction section, concentrating on sustainable & eco-friendly fashion in Vietnam. We hope the new version is more concise, convincing, and relevant.

3) In the LITERATURE REVIEW, the authors could include a better explanation of the loyalty and fashion sector. In this sense, the authors could include some relevant and references from the fashion sector such as:

Vinhas Da Silva, R. and Faridah Syed Alwi, S. (2006), "Cognitive, affective attributes and conative, behavioural responses in retail corporate branding", Journal of Product & Brand Management, Vol. 15 No. 5, pp. 293-305. https://doi.org/10.1108/10610420610685703

Kuikka, A. and Laukkanen, T. (2012), "Brand loyalty and the role of hedonic value", Journal of Product & Brand Management, Vol. 21 No. 7, pp. 529-537. https://doi.org/10.1108/10610421211276277

Cuesta-Valiño, P., Gutiérrez-Rodríguez, P. and Núnez-Barriopedro, E. (2021), "The role of consumer happiness in brand loyalty: a model of the satisfaction and brand image in fashion", Corporate Governance, Vol. ahead-of-print No. ahead-of-print. https://doi.org/10.1108/CG-03-2021-0099

Dear reviewer,

We have adopted new citations and papers from your suggestions and the Vietnamese and global literature. We hope that the new information reflects our literature review more nuancedly. Since we re-construct our literature substantially, we are open to receiving more suggestions from you.

4) In the METHOS section, the authors could explain better all the steps carried out in the process of obtaining the data for this investigation.

Dear reviewers,

We have added the following information for the quantitative section for data collection:

The authors used convenience and snowball sampling to recruit generation Z residents in Vietnam through a Qualtrics survey. These sampling techniques help researchers target the right participant group and achieve the sample size threshold in a short period of time (August to September 2021) with an affordable budget. We distributed the Qualtrics questionnaire on social media platforms, namely Facebook, on researchers’ accounts in a public setting. We also posted on various Facebook groups in high school and university student communities and young professional communities to recruit more respondents.

Why have only 24 people been selected? For example, why are there 6 males and 17 females?

Dear reviewer,

The qualitative phase is a pilot study for exploratory purposes rather than generalizing knowledge. We only aim to understand young customers' perception and incorporate it with the literature to construct the conceptual models. Hence, we only ensure the most important factor, the age of participants, rather than their genders. The knowledge generalization is in the quantitative section (SEM-PLS).

Is the selected sample (313) representative and enough? Are these people representative of the population the authors want to study? Why are there so many more women than men?

Dear reviewers,

We distributed the surveys at university & high school student Facebook groups on social media due to the Covid-19 pandemic. As we are looking for young customers, our data reflect this population. Regarding the difference between male & female respondents, we have used MGA-PLS analysis to identify whether the two groups are significantly different. The result shows no significant difference in the relationships between constructs in the models & their paths between the genders of respondents in this case. We have added this information to the manuscript. Regarding the reason there are more female respondents compared to male respondents, we believe that young female customers care more about fashion than male customers. Since we are using convenience and snowball sampling, the current sample might reflect this nature.

Path Coefficients-diff (Female - Male)

p-Value original 1-tailed (Female vs Male)

p-Value new (Female vs Male)

CF -> CS

0.043

0.368

0.736

CS -> CL

0.083

0.254

0.508

CS -> PI

0.052

0.343

0.685

Envf -> CL

0.158

0.069

0.138

Envf -> PI

-0.255

0.973

0.054

PBC -> CL

-0.18

0.936

0.129

PBC -> PI

0.17

0.085

0.171

PVI -> CS

-0.121

0.85

0.3

PI -> CL

-0.04

0.637

0.726

The authors should explain better why they have used the SEM-PLS for the investigation.

Dear reviewer,

We have added the information to justify the use of SEM-PLS:

The quantitative phase explores whether the relationships between variables in the conceptual model are statistically significant (Figure 1). The analysis also identifies the factors that have the strongest impact on the purchase intention of sustainable fashion products and their impact on customer loyalty. As the existing literature has mentioned the strength of the SEM-PLS technique in exploratory or an extension of an existing structural theory compared to CB-SEM, this research employs the SEM-PLS technique [61]. Second, this method is useful for the prediction of relationships and is more suitable for theoretical development than other methods, such as CB-SEM. This is important because the current research model is not heavily based on well-established theories. Third, this technique also allows analysis with small sample size and has no assumptions regarding data distribution [61]. Finally, as customers’ perceptions are very subjective and there are differences between cultures, this method is rigorous in analysis and easy to replicate in future research in another context.

5) The DISCUSSIONS section is very poor. Therefore, the authors should improve much this section on theoretical implications and practical implications, and they should end the section with the limitations of their study and the most important conclusion of their research.

Dear reviewer,

We have revised our discussion section and followed your structure. We also added the information on how this research gives new insight into the Vietnamese consumption culture, at least in the field of eco-friendly fashion. Please see our new discussion section for more information.

Quantitative analysis shows that while PBC has a moderate effect on purchase intention; environmental concern and customer satisfaction have only a small effect. This section sheds light on this relationship. First, the Vietnamese generation Z may have knowledge and attitude about the environment, but do not have enough resources or motivation to buy eco-friendly products. The main barrier faced by most respondents in the qualitative phase was the lack of available eco-friendly products; hence, only customers who truly had spare time and financial resources were willing to buy these products. Many interviewees expressed that their lack of eco-friendly fashion consumption can be traced back to their lack of access to local eco-friendly shops and brands. In addition, customer-driven factors, namely environmental concerns, and PBC do not significantly affect loyalty. This relationship is evident in both qualitative and quantitative phrases. These insignificant results contradict the findings in previous studies . This qualitative finding also implies that Vietnamese youth do not automatically associate eco-friendly fashions with high-quality products, which is contrary to the findings in previous study [21].

Moreover, the international and Vietnam-based studies from producers' perspectives identify five major elements influencing customers' purchase intention of eco-friendly products: transparency, influencers, acculturation, and self-esteem [71,72]. Our study found only 3 of these elements, and no Gen Z participants mentioned acculturation. The difference between customers' perceptions in this study and producers' perceptions in previous studies highlights the gap between these stakeholders. Moreover, many previous studies have assumed that Vietnamese consumers have collectivist cultures. However, at least in our findings on the eco-friendly fashion industry, Generation Z participants illustrated individualism in both quantitative and qualitative results. Our participants concentrated on product quality (PIV), self-esteem, and resources (CPB) in making decisions rather than their knowledge of environmental consequences (Enfv). This new finding might be explained by the growth of Vietnamese Generation Z in a peaceful and modernized country without worrying about survivability - the pressure that previous generations must collectively join hands to achieve. With this contrasting finding from the literature, businesses and scholars should facilitate more research on generation Z to capture their unique nature, preventing stereotyping and outdated data from hampering effective business and marketing decisions.

Notably in the qualitative section, most interviewees suggested that firms should raise awareness and increase the accessibility of eco-friendly products to promote sustainable consumption. However, the SEM-PLS results illustrate that Gen Z’s knowledge and awareness are insignificant compared to perceived product quality and control behaviors. Hence, the contradiction of Gen Z’s suggestions in the qualitative phase and their responses in the quantitative phase implies that Gen Z’s perceptions might not be consistent with their actions. This contradiction may also imply that the qualitative phase is vulnerable to the social desirability bias. Future research should investigate their behavior from more rigorous psychological methods.

Reviewer #2: Thank you for the opportunity to review this paper. The research was very interesting particularly through the cultural lens of Vietnam. The introduction section successfully introduces the background context to which the research resides. Whilst this section was well written and the writing style has a nice flow I would have liked to have been given a sense of purpose at the beginning of the introduction section. This would have included locating the study in Vietnam.

Dear Reviewer 2,

Thank you very much for your comments. It is very constructive, and we have made the necessary changes according to your suggestions. We hope to hear more feedback from you and thank you for your support. 

The introduction covers policy and legislation from international governments, businesses again with an international approach, corporate social responsibility, and consumer behaviour. Collectively, this leads to a lack of clarity Regarding which literature stream this paper contributes to. Direction of the paper should be clearer by this stage.

In the newest version of the Introduction, we have truncated the non-management theories and information you suggested. We also re-organized our introduction section, which more concentrate on sustainable & eco-friendly fashion in Vietnam. We hope the new version is more concise, convincing, and relevant.

It would be good to set the cultural context of Vietnam, there are discussions around behaviours and markets in global contexts and it would be good to know how this directs the research in Vietnam and the implications this has for the findings. I would have liked to have known what the fashion retail environment within Vietnam was like, for example where there are many international brands or Vietnamese brands are there large chain stores or are there smaller boutiques a better understanding of this would help to understand the whole context of the paper.

Dear reviewers, 

We also highlight the Vietnamese fashion, garment, and textile markets, illustrating their domestic significance and roles as a global export in the global supply chain. The peer-reviewed articles on the Vietnamese eco-friend/sustainable fashion and garment market are very limited; hence, we have incorporated some corporate reports & news from reliable sources. We also discuss the previous research on Vietnamese and explain why its favorable culture toward sustainable products has not been utilized fully.

At the end of our introduction section, we have stated clearly our research question:

 RQ1: Which factors have a stronger impact on consumer loyalty and purchase intention toward eco-friendly fashion in Vietnam: developing quality eco-friendly products and services or promoting consumers' understanding of environmental preservation?

 RQ2: From the interviewees’ perspective, how to encourage the consumption of eco-friendly products in Vietnam?

Additionally, I wonder what the link is between the variables in Figure 1 and sustainability, this could be clearer. To address generation z Vietnam consumer's opinions on sustainability and the fashion industry is highly relevant and makes a valid contribution to the literature. However, this could be more strongly linked to the idea about post purchase satisfaction and the other variables mentioned. Focused research questions at the end of the introduction would help to better understand what the research seeks to investigate. This sense of purpose would also help positioning of the literature review and this could be further established within the development of the literature reviewed. Clearer clarification on what the research is investigating would be helpful. The literature review begins also by covering a number of topics and would benefit from focusing more acutely on the research agenda. Links between sustainable fashion and post consumption satisfaction from other contexts could be more clearly articulated within the literature review.

Dear reviewer,

We have adopted new citations and papers from your suggestions and the Vietnamese and global literature. We hope that the new information reflects our literature review more nuancedly.

Often times references were missing to support points made and knowledge development. Additionally, a lot of research around sustainable fashion consumption behaviours, particularly, using the TRA was not included within the literature discussion. deepening this discussion would have helped create a stronger theoretical construct. I would have liked to have seen a more developed literature review that focused specifically on sustainable fashion consumption behaviours and what is already known about them.

Dear Reviewer,

In this newest Literature Review version, we have directly entered sustainable fashion from business perspectives & its theoretical framework. We started to explore previous literature, and we realized most of them are from TRA and TBP. On the theoretical contribution, as mentioned in the introduction, the literature examines the customer's environmental concern as a factor rather than customers' perceived satisfaction with product and service quality. In short, they are concentrating only on the Promotion elements of the marketing mix (4P). Therefore, rather than only investigating the environmental concerns and environmental awareness and perception like previous TRA and TPB studies, we adopted a second set of constructs illustrating the product & service qualities and price value of the eco-friendly fashion products, addressing the additional two marketing elements (Product and Price). From a practical perspective, rising environmental awareness and product development are two separate and costly investments that many small and medium-sized businesses cannot afford. Therefore, the literature gaps motivate us to find which factors have a stronger impact on consumer loyalty and purchase intention toward eco-friendly fashion: consumers' understanding of environmental preservation or the development of quality eco-friendly products and services? This research question pays the foundation for constructing the conceptual framework and research hypotheses.

p. 6 According to [24-26] - should this have the authors names? I found this a few times. Such as on page 11: According to [54],

Dear reviewer,

You were correct. We have consulted the PLOS One format and re-edited these typos. Thank you for pointing out our mistake in the formatting.

I especially like to the table in which the data analysis steps were presented undescribed. The methodology section was well explained offering transparency on how data were collected. I would have liked to have known how the quantitative sample were accessed and communicated with.

Dear reviewers,

We have added the following information for the quantitative section for data collection:

The authors used convenience and snowball sampling to recruit generation Z residents in Vietnam through a Qualtrics survey. These sampling techniques help researchers target the right participant group and achieve the sample size threshold in a short period of time (August to September 2021) with an affordable budget. We distributed the Qualtrics questionnaire on social media platforms, namely Facebook, on researchers’ accounts in a public setting. We also posted on various Facebook groups in high school and university student communities and young professional communities to recruit more respondents. 

Other information has been reported on the COREQ statement in our Appendix.

The qualitative data was presented in a quantitative manner and this did not communicate the perceived rich data that is typical of a qualitative study. There were no participant quotes presented in support of analytic development.

Dear Reviewer,

We have incorporated quotes in our qualitative analysis. All transcripted data is also available in public for future reference at: https://doi.org/10.6084/m9.figshare.17021879.v2

I liked that the qualitative data was used to further to develop hypothesis and this helps to tighten up the methodological approach.

Thank you for your comments.

The data from the qualitative study is highly similar to the consumer behaviour sustainable fashion literature that has existed for the last 20 years. It would be good to have seen some kind of advancement on previous knowledge and perhaps it could have come from a cultural aspect.

Dear reviewer,

We have revised our discussion section and followed your structure. We also added the information on how this research gives new insight into the Vietnamese consumption culture, at least in the field of eco-friendly fashion:

Quantitative analysis shows that while PBC has a moderate effect on purchase intention; environmental concern and customer satisfaction have only a small effect. This section sheds light on this relationship. First, the Vietnamese generation Z may have knowledge and attitude about the environment, but do not have enough resources or motivation to buy eco-friendly products. The main barrier faced by most respondents in the qualitative phase was the lack of available eco-friendly products; hence, only customers who truly had spare time and financial resources were willing to buy these products. Many interviewees expressed that their lack of eco-friendly fashion consumption can be traced back to their lack of access to local eco-friendly shops and brands. In addition, customer-driven factors, namely environmental concerns, and PBC do not significantly affect loyalty. This relationship is evident in both qualitative and quantitative phrases. These insignificant results contradict the findings in previous studies . This qualitative finding also implies that Vietnamese youth do not automatically associate eco-friendly fashions with high-quality products, which is contrary to the findings in previous study [21].

Moreover, the international and Vietnam-based studies from producers' perspectives identify five major elements influencing customers' purchase intention of eco-friendly products: transparency, influencers, acculturation, and self-esteem [71,72]. Our study found only 3 of these elements, and no Gen Z participants mentioned acculturation. The difference between customers' perceptions in this study and producers' perceptions in previous studies highlights the gap between these stakeholders. Moreover, many previous studies have assumed that Vietnamese consumers have collectivist cultures. However, at least in our findings on the eco-friendly fashion industry, Generation Z participants illustrated individualism in both quantitative and qualitative results. Our participants concentrated on product quality (PIV), self-esteem, and resources (CPB) in making decisions rather than their knowledge of environmental consequences (Enfv). This new finding might be explained by the growth of Vietnamese Generation Z in a peaceful and modernized country without worrying about survivability - the pressure that previous generations must collectively join hands to achieve. With this contrasting finding from the literature, businesses and scholars should facilitate more research on generation Z to capture their unique nature, preventing stereotyping and outdated data from hampering effective business and marketing decisions.

Notably in the qualitative section, most interviewees suggested that firms should raise awareness and increase the accessibility of eco-friendly products to promote sustainable consumption. However, the SEM-PLS results illustrate that Gen Z’s knowledge and awareness are insignificant compared to perceived product quality and control behaviors. Hence, the contradiction of Gen Z’s suggestions in the qualitative phase and their responses in the quantitative phase implies that Gen Z’s perceptions might not be consistent with their actions. This contradiction may also imply that the qualitative phase is vulnerable to the social desirability bias. Future research should investigate their behavior from more rigorous psychological methods.

Overall I would have liked to see the strong consistency within this paper setting out clear parameters that would be under investigation and link to the conclusions and discussion.

Dear Reviewer,

We hope the current version has been more consistent overall.

---

## [Decision Letter · Decision Letter 1]

27 Jul 2022

Eco-friendly Fashion among Generation Z: Mixed-methods study on Price Value Image, Customer Fulfillment, and Pro-environmental Behavior

PONE-D-21-36570R1

Dear Dr. Nguyen,

We’re pleased to inform you that your manuscript has been judged scientifically suitable for publication and will be formally accepted for publication once it meets all outstanding technical requirements.

Kind regards,

Maurizio Naldi

Academic Editor

PLOS ONE

Additional Editor Comments (optional):

Reviewers' comments:

Reviewer's Responses to Questions

**Comments to the Author**

1. If the authors have adequately addressed your comments raised in a previous round of review and you feel that this manuscript is now acceptable for publication, you may indicate that here to bypass the “Comments to the Author” section, enter your conflict of interest statement in the “Confidential to Editor” section, and submit your "Accept" recommendation.

Reviewer #1: All comments have been addressed

2. Is the manuscript technically sound, and do the data support the conclusions?

Reviewer #1: Yes

3. Has the statistical analysis been performed appropriately and rigorously? 

Reviewer #1: Yes

4. Have the authors made all data underlying the findings in their manuscript fully available?

Reviewer #1: Yes

5. Is the manuscript presented in an intelligible fashion and written in standard English?

Reviewer #1: Yes

6. Review Comments to the Author

Reviewer #1: Dear authors,

First of all, I am glad to have the opportunity to read again the article on “Promoting Eco-friendly Fashion among Youth Generation in Developing Countries: Mixed-methods study on The Roles of Price Value Image, Customer Fulfillment, and Pro-environmental Behavior”, that I have read with great interest.

The authors have done a great job with the improvements made to the manuscript, so they have improved in quality. For this reason, I recommend the publication in its present form. Congratulations

7. PLOS authors have the option to publish the peer review history of their article (what does this mean?). If published, this will include your full peer review and any attached files.

Reviewer #1: No

---

## [Editor Report · Acceptance letter]

5 Aug 2022

PONE-D-21-36570R1 

Eco-friendly Fashion among Generation Z: Mixed-methods study on Price Value Image, Customer Fulfillment, and Pro-environmental Behavior 

Dear Dr. Nguyen:

I'm pleased to inform you that your manuscript has been deemed suitable for publication in PLOS ONE. Congratulations! Your manuscript is now with our production department. 

Kind regards, 

on behalf of

Professor Maurizio Naldi 

Academic Editor

PLOS ONE